# Numerical Simulation Analysis of Fracture Propagation in Rock Based on Smooth Particle Hydrodynamics

**DOI:** 10.3390/ma16196560

**Published:** 2023-10-05

**Authors:** Xuhua Ren, Hui Zhang, Jixun Zhang, Shuyang Yu, Semaierjiang Maimaitiyusupu

**Affiliations:** 1College of Water Conservancy and Hydropower Engineering, Hohai University, Nanjing 210024, China; renxh@hhu.edu.cn (X.R.); zhangh1998hhu@163.com (H.Z.); samarjan@hhu.edu.cn (S.M.); 2School of Transportation and Civil Engineering, Nantong University, Nantong 226019, China; yushuyang_hhu@163.com

**Keywords:** rock, crack propagation, failure characteristics, SPH

## Abstract

The mechanical properties of fractured rock have always been a focal point in the rock mechanics field. Based on previous research, this paper proposes improvements to the SPH method and applies it to the study of crack propagation in fractured rocks. By conducting uniaxial compression tests and simulating crack propagation on various specimens with different crack shapes, the characteristics of crack propagation were obtained. The comparison between the simulated results in this study and existing experimental and numerical simulation results confirms the validity of the SPH method employed in this paper. The present study utilizes the proposed methodology to analyze the influence of the crack angle, width, and orientation on crack propagation. The SPH method employed in this study effectively demonstrates the expansion process of fractured rock under uniaxial compression, providing valuable insights for the engineering applications of SPH.

## 1. Introduction

Under the influence of geological mechanics, rock formations often harbor inherent imperfections, such as microfractures, fissures, cavities, and faults, among others. The mechanical characteristics of rock formations are significantly influenced by these inherent imperfections, impacting factors such as deformability, strength, and failure patterns, among others [1,2]. The locations where fractures exist often exhibit poor uniformity, nonlinearity, and discontinuity, making them prone to stress concentration, consequently resulting in the brittle failure of the rock formation, thereby affecting the overall mechanical performance of the rock [3]. Therefore, studying the propagation of cracks in rocks is of particular significance.

Currently, the main methods for studying the propagation of cracks in rocks are experimental testing and numerical simulation. Tang conducted uniaxial compression tests on granite specimens with different pre-existing defects, and found that the crack in the granite specimen would undergo localized deformation and failure [4]. Yang and others conducted compression tests on sandstone specimens with three non-continuous cracks to study the process of crack propagation, coalescence, and rock failure [5]. Sun conducted uniaxial compression tests on square shale and rectangular sandstone with multiple cracks at different relative positions and depositional angles, verifying the criteria for crack initiation in anisotropic rocks [6]. However, due to the complexity of specimen preparation and the application of boundary conditions, numerical simulation methods have become increasingly adopted by scholars. These methods require the input of geometric parameters, mechanical parameters, and boundary conditions, and can provide good results [7].

Currently, the finite element method, discrete element method, and numerical manifold method are widely used numerical simulation techniques. The finite element method is also an early numerical simulation method for studying the expansion of cracks in rock mechanics. Since the finite element method discretizes the structure into a model composed of a finite number of elements and the expansion of cracks is unknown, this approach requires the redivision of elements after each calculation [8,9]. Although the mesh refinement technique in the finite element method can address the issue of an unknown crack propagation direction, it involves a significant amount of work. Moreover, due to substantial geometric variations near the cracks, ensuring the quality of the elements becomes challenging. This can potentially lead to non-convergent computational results or significant discrepancies with actual errors [9,10]. Subsequently, a specialized finite element method for computing crack propagation was proposed and widely adopted. The main idea of this approach is to represent the shape of the crack using a specific mathematical function [11,12]. However, the extended finite element method is incapable of handling scenarios involving triangular or pentagonal models, or intersecting cracks. The discrete element method is a numerical simulation technique that originated from the development of the finite element method. It employs an unstructured grid approach and utilizes various contact interactions between particles to simulate the process of crack initiation and propagation [13,14]. Kim proposed a finite element method with local mesh refinement at the crack tip, which is highly effective for simulating crack propagation in shell structures [15,16]. Compared to the finite element method, the discrete element method offers the advantage of circumventing the need for grid partitioning, making it more suitable for simulating the initiation, propagation, and interaction of cracks. However, the discrete element method requires meticulous determination of the conversion of macroscopic parameters into microscopic ones, which entails a higher demand for parameter specifications [17].

In recent years, researchers have been applying novel numerical simulation methods to the calculation of rock fracture propagation. Zhou proposed a micro-elastic/plastic constitutive model that conforms to the mechanical characteristics of rock materials, and introduced the Weibull distribution function to reflect the heterogeneity of rocks [18]. The numerical manifold method employs two sets of independent mathematical and physical grids to solve both continuous and discontinuous problems in a unified manner. The discontinuous deformation analysis method describes the motion of individual blocks using Newton’s laws, without allowing them to embed into each other. By establishing a system of overall equilibrium equations based on the principle of minimum potential energy, the method implicitly solves for the displacement of each block [19,20]. Smooth Particle Hydrodynamics (SPH) is an immaculate Lagrangian mesh-free particle algorithm that is exceptionally well-suited for addressing profound deformations and discontinuities in problem domains. Consequently, in recent years, it has begun to find application in the realm of rock fracture mechanics [21]. Following the groundwork laid by SPH, Zhou introduced a novel concept known as Generalized Particle Dynamics (GPD), which has subsequently found significant utility in the field of rock fracture mechanics [22,23,24,25,26,27]. Building upon this foundation, Yu proposed an Improved Kernel of Smooth Particle Hydrodynamics (IKSPH) by altering the kernel function. This innovative approach has successfully simulated the process of crack propagation in fractured materials under compressive states, as well as the failure mechanism of slopes featuring jointed fractures [28,29,30,31,32]. Mu proposed an SPH method that considers the coupling of water–kinetic damage. This method is used to simulate the crack propagation and merging processes in rocks with prefabricated defects under compression and the influence of defect water pressure [33].

Due to the intrinsic virtue of the SPH methodology, the necessity of constructing a grid is obviated, thereby circumventing quandaries associated with grid distortion. Furthermore, its remarkable aptitude for accommodating substantial deformation and discontinuities must be underscored. This paper proposes a numerical simulation method based on fracture mechanics and SPH, which can effectively simulate the fracturing process of cracked rock. Using this method, the load tests on several typical fractured specimens were simulated, and the crack propagation behavior of these specimens was obtained. The simulation results were compared with existing experiments or numerical simulations to verify the accuracy of this study. Finally, the method was used to simulate specimens with different inclination angles, widths, and lengths of cracks, revealing the relationship between crack propagation and the crack inclination angle, width, and length. The identified patterns from this research provide references for rock slope and tunnel engineering and have practical implications for support measures.

## 2. The Basic Theory of SPH

### 2.1. Basic Ideas of SPH

The SPH method was devised with the purpose of addressing the partial differential equations in fluid dynamics, revolving around the approximation of kernel functions and particle-based discretization.

### 2.2. SPH Basic Formula

The numerical computation of SPH primarily involves two steps: the first step being the integral representation method, followed by the particle approximation method [24,25,26,27,28]. In SPH, the fundamental equations are expressed through integral representation, as follows:(1)$$f\left(x\right)=\int_{\Omega }f\left(x^{\prime }\right)\delta \left(x-x^{\prime }\right){dx}^{\prime }$$
where x is a three-dimensional coordinate vector; $x^{\prime }$ is the three-dimensional coordinate vector of a specific point; f(x) is the field function about x; and $\delta (x-x^{\prime })$ is the Dirac function, and its properties can be expressed as
(2)$$\delta (x-dx^{\prime })=\left\{\begin{array}{l} 1,x=x^{\prime }\\ 0,x\neq x^{\prime } \end{array}\right.$$

If smoothness $W(x-x^{\prime },h)$ is used instead of the Dirac function, its field function is expressed as
(3)$$f\left(x\right)=\int_{\Omega }f\left(x^{\prime }\right)W(x-x^{\prime },h){dx}^{\prime }$$

Replacing $f\left(x\right)$ with spatial derivative $\nabla •f\left(x\right)$ yields
(4)$$\nabla •f\left(x\right)\approx \int_{\Omega }\left[\nabla •f\left(x^{\prime }\right)\right]W(x-x^{\prime },h){dx}^{\prime }$$

By using the divergence theorem to transform the right side of Equation (4), it can be found that
(5)$$\nabla •f\left(x\right)=\int_{S}f\left(x^{\prime }\right)W(x-x^{\prime },h)•{ndx}^{\prime }-\int_{\Omega }f\left(x^{\prime }\right)•\nabla W(x-x^{\prime },h){dx}^{\prime }$$

Due to the compact support property of the smooth function (where its values outside the support domain are zero), the value of $\int_{S}f\left(x^{\prime }\right)W(x-x^{\prime },h)•{ndx}^{\prime }$ is zero. Therefore, the equation can be elegantly condensed as
(6)$$\nabla •f\left(x\right)=-\int_{\Omega }f\left(x^{\prime }\right)•\nabla W(x-x^{\prime },h){dx}^{\prime }$$

In the Smoothed Particle Hydrodynamics (SPH) method, the particles can be regarded as having independent masses and occupying distinct regions in space. The mass of a particle can be denoted as
(7)$$m_{j}=\Delta V_{j}\rho_{j}$$
where $\rho_{j}$ is the density of particle j(j=1,2,…,N), N is the total number of particles within the support domain of particle j, and $\Delta V_{j}$ is the volume of the voxel at particle j [27,28].

The derivative approximation formula of the field function at particle i can be derived through the transformation of Equation (6):(8)$$\begin{array}{ll} \nabla •f\left(x_{i}\right) & =-\int_{\Omega }f\left(x^{\prime }\right)•\nabla W(x-x^{\prime },h){dx}^{\prime }\\ & =-\sum_{j=1}^{N}\frac{m_{j}}{\rho_{j}}f\left(x_{j}\right)\nabla W_{ij} \end{array}$$

In Equation (8), we introduce the density into the gradient operator
(9)$$\nabla •f\left(x\right)=\rho \left[\nabla •\left(\frac{f\left(x\right)}{\rho }\right)-\frac{f\left(x\right)}{\rho^{2}}•\nabla \rho \right]$$

By calculating the values of $\nabla •\left(\frac{f\left(x\right)}{\rho }\right)$ and ∇ρ in Equation (9) using Equation (8), the approximate expression for the particle can be obtained as follows:(10)$$\nabla •f\left(x_{i}\right)=\rho_{i}\left[\sum_{j=1}^{N}m_{j}\left[\frac{f\left(x_{j}\right)}{\rho_{j}^{2}}-\frac{f\left(x_{i}\right)}{\rho_{i}^{2}}\right]•\nabla_{i}W_{ij}\right]$$

### 2.3. Smooth Function

The utilization of smooth functions plays a crucial role in the computation of SPH, significantly impacting both the precision and efficiency of calculations [24,25,26,27,28]. A myriad of conventional smooth functions exist, such as the bell-shaped functions, Gaussian functions, and higher-order spline functions, among others. This article adopts a segmented quintic spline function:(11)$$W(R,h)=\alpha_{d}•\left\{\begin{array}{c} {\left(3-R\right)}^{5}-6{\left(2-R\right)}^{5}+15{\left(1-R\right)}^{5}\;\;\;0\leq R<1\\ {\left(3-R\right)}^{5}-6{\left(2-R\right)}^{5}\;\;\;\;\;\;1\leq R<2\\ {\left(3-R\right)}^{5}\;\;\;\;\;\;\;\;\;\;\;\;2\leq R\leq 3 \end{array}\right.$$
where $\alpha_{d}$ is taken as 1/120 h, $7/\left(478\;\pi \;h^{2}\right)$, and $1/\left(120\;\pi \;h^{3}\right)$ when the problem is one-dimensional, two-dimensional, and three-dimensional, respectively. R is the ratio of particle spacing to smooth length.

In contrast to using third-order smooth functions extensively, the utilization of fifth-order smooth functions in this paper possesses the following characteristics: (1) The value of the smoothing coefficient being smaller leads to a higher level of precision in the solution, rendering it more accurate and aptly aligned with real-world circumstances. (2) The curve of the second derivative exhibits enhanced smoothness. Given that the stability of SPH calculations is significantly influenced by the second derivative of the smoothing function, the adoption of a fifth-order smooth function in this study ensures superior stability. (3) Its continuity is better.

## 3. Implementation Method

### 3.1. Particle Search Method

In the context of the SPH method, due to the finite-range compact support domain of the smoothing function, the particle’s support domain only contains a limited number of particles. These particles within the support domain are referred to as neighboring particles (NNP). The most straightforward method to search for neighboring particles around each particle is to compute the distance between the given particle and all other particles. Any particle for which the distance from the given particle is less than the radius of the support domain is considered an adjacent particle. Although this approach is relatively simple to implement, it involves a high computational cost due to the large number of calculations required. Specifically, its computational complexity is O(n^2^), where n represents the number of particles. Consequently, as the number of particles increases, the computational burden becomes exceedingly significant. Due to the high number of search operations involved in the direct search method, the linked list approach has been proposed as an alternative. This method divides the computational domain into several grids, as shown in Figure 1. The size of each grid is determined based on the size of the support domain. Each particle’s search range is limited to the grid it belongs to or its neighboring grids. By employing a simple storage mechanism, particles within each grid are efficiently stored. The computational complexity of this method remains O(n^2^) [28].

The search region in the two-dimensional linked list search method is depicted as shown in Figure 1. For a specific particle, its search range is limited to the 9 neighboring grids only. If the computational scope is limited to the currently displayed 25 grids, the search scope in the two-dimensional linked list search approach constitutes 9/25 of that in the direct search approach. In practice, the actual computational scope exceeds the current number of grids. Hence, it can be concluded that the computational efficiency of the linked list search method far surpasses that of the direct search method in practical scenarios.

### 3.2. Boundary Condition

In the context of SPH calculations, the presence of significant flaws in particles located at or near the boundaries leads to an incomplete influence domain for particles on the boundary. As suggested in the literature, there are two types of virtual particles. The first type of virtual particle is set on the boundary. The second type of virtual particle is arranged on the boundary and is symmetrically positioned with respect to the real particles. These virtual particles and real particles share the same density and pressure if virtual particles with the same magnitude but opposite direction of velocity are arranged, as shown in Figure 2. Due to the dynamic nature of real particles, it is necessary to regenerate the virtual particles based on the current state of the real particles at each computational time step [25,26,27,28,29,30,31].

### 3.3. Time Integral

The time integration of the SPH method employed in this text utilizes the leapfrog approach [26,28].
(12)$$\left\{\begin{array}{l} \rho_{i}(t_{0}+\frac{\Delta t}{2})=\rho_{i}(t_{0})+\frac{\Delta t}{2}d\rho_{i}(t_{0})\\ v_{i}(t_{0}+\frac{\Delta t}{2})=v_{i}(t_{0})+\frac{\Delta t}{2}dv_{i}(t_{0})\\ x_{i}(t_{0}+\frac{\Delta t}{2})=x_{i}(t_{0})+\frac{\Delta t}{2}dx_{i}(t_{0}) \end{array}\right.$$
where $t_{0}$ is the current moment and Δt is the time increment.

### 3.4. Realization of Solid Mechanics Constitutive Equation in SPH

The governing equations for the solid mechanics SPH method are based on the conservation equations of continuum mechanics, which include the equations of mass conservation, momentum conservation, and energy conservation [29,30]. The expressions for these equations are as follows:(13)$$\frac{d\rho }{dt}=-\rho \frac{\partial v^{\beta }}{\partial x^{\beta }}$$
(14)$$\frac{dv^{\alpha }}{dt}=\frac{1}{\rho }\frac{\partial \sigma^{\alpha \beta }}{\partial x^{\beta }}$$
(15)$$\frac{de}{dt}=\frac{\sigma^{\alpha \beta }}{\rho }\frac{\partial v^{\alpha }}{\partial x^{\beta }}$$
where ρ is the density, e is the internal energy, $v^{\alpha }$ is the velocity component, $\sigma^{\alpha \beta }$ is the total stress component, t is the time, and x is the location. x and t are independent variables.

After transformation, it can be obtained that
(16)$$\frac{d\rho_{i}}{dt}=\sum_{j=1}^{N}m_{j}v_{ij}^{\beta }\frac{\partial W_{ij}}{\partial x_{i}^{\beta }}$$
(17)$$\frac{dv_{i}^{\alpha }}{dt}=\sum_{j=1}^{N}m_{j}\frac{\sigma_{i}^{\alpha \beta }+\sigma_{j}^{\alpha \beta }}{\rho_{i}\rho_{j}}\frac{\partial W_{ij}}{\partial x_{i}^{\beta }}$$
(18)$$\frac{de_{i}}{dt}=\frac{\sigma_{i}^{\alpha \beta }}{\rho_{i}^{2}}\sum_{j=1}^{N}m_{j}(v_{j}^{\beta }-v_{i}^{\beta })W_{ij,\beta }$$

For the constitutive equations of a solid, stress is expressed as a function of strain and strain rate, and can be represented by the following equation:(19)$${\bar{\tau }}^{\alpha \beta }=B\left(\varepsilon^{\alpha \beta }-\frac{1}{3}\delta^{\alpha \beta }\varepsilon^{rr}\right)+\tau^{\alpha r}R^{\beta r}+\tau^{r\beta }R^{\alpha r}$$
where $\bar{\tau }$ is the stress rate tensor, B is the shear modulus, and R is the torsion tensor; its expression is
(20)$$R^{\alpha \beta }=\frac{1}{2}\left(\frac{\partial v^{\alpha }}{\partial x^{\beta }}-\frac{\partial v^{\beta }}{\partial x^{\alpha }}\right)$$

### 3.5. Brittle Damage Fracture Model

In most cases, rock degradation is attributed to the occurrence of brittle fractures within the rock’s interior, leading to destruction. Employing a model that encompasses the concepts of elasticity and brittleness will yield a more accurate simulation of material fracture [28]. This paper adopts an elastic–brittle constitutive model to simulate the process of rock fracture. The particles adhere to the principle of maximum tensile stress and Mohr–Coulomb criterion:(21)$$\sigma_{f}=\sigma_{t}$$
(22)$$\tau_{f}=c+\sigma_{f}\tan\varphi$$
where $\sigma_{f}$ is the maximum tensile stress on the failure surface, $\sigma_{t}$ is the material’s tensile strength, $\tau_{f}$ is the maximum shear stress on the failure surface, c is the cohesion of the material, and φ is the internal friction angle. When particles satisfy both tensile failure and shear failure simultaneously, it is preferable to classify the condition as tensile failure.

This calculation introduces a fracture coefficient, denoted as D, for each particle. The fracture mechanism of the particle is determined by comparing the shear or tensile stress it experiences with the allowable values defined in Equations (21) or (22). If the particle satisfies either condition in Equations (21) or (22), its fracture coefficient D is set to 0. Particles with a fracture coefficient of 0 are identified as damaged particles and are excluded from force transmission in subsequent calculations. This setup aims to simulate the fracture behavior of materials under loading conditions and analyze material stability.

### 3.6. Programming Architecture

The program architecture of this paper is shown in Figure 3, and the calculation process is as follows: (1) Firstly, input the geometric information of the model, the number of calculation steps, and other mechanical parameters. Then, generate internal particles based on the aforementioned conditions. (2) Based on the boundary conditions, generate two types of boundary particles. (3) Utilize the linked list search method to conduct a search on the particles. (4) Perform density calculations. (5) Update the fracture coefficient based on the damage information. (6) Calculate the internal force. (7) Calculate the artificial thermal energy and viscosity. (8) Update speed. (9) Output the computed results. (10) Determine whether the desired number of computation steps has been reached. If it has, end the calculation. If not, return to step (2) and continue looping until the desired number of computation steps is achieved.

## 4. Example Analysis

### 4.1. Example Size

To substantiate the rationality of the present SPH method utilized in this text, several distinct compression cases are employed as examples. They will be compared to existing experimental or numerical simulation results for validation purposes. The five computational examples consist of specimens with inclined cracks, intersecting cracks, double V-shaped cracks, circular holes, and circular holes with inclined cracks. The dimensions and models of these examples are depicted in Figure 4 and Figure 5. The cohesive strength of the specimens is set to 5.0 MPa, the internal friction angle is set to 40°, the tensile strength is set to 1.0 MPa, and the loading rate is 5 mm/s. The time step size for each individual step is 10^−8^ m.

### 4.2. Calculation Results

The calculation results of the five cases are shown in Figure 6, Figure 7, Figure 8, Figure 9 and Figure 10. These four contour maps illustrate the distribution of maximum principal stress at different analysis steps (1000, 2000, 3000, and 6000 steps), corresponding to displacement loads of 0.01 mm, 0.02 mm, 0.03 mm, and 0.06 mm. The stress is measured in units of Pa, where positive values represent tensile stress.

The computation results of example 1 are shown in Figure 6, and a total of 11,216 particles were used in this example. When the displacement is loaded to 0.01 mm, the maximum tensile stress occurs on both sides of the crack, with noticeable compressive stress on the crack surfaces. When the displacement is loaded to 0.02 mm, stress concentration occurs on both sides of the crack. When the displacement is loaded to 0.03 mm, the stress concentration region expands upwards and downwards, and the width of the high-stress area also increases. When the displacement is loaded to 0.06 mm, the wing-shaped stress concentration area further expands in both upward and downward directions, and its width increases accordingly. In subsequent loading, the range of the stress concentration area no longer increases significantly, and the specimen experiences failure.

The calculation results of example 2 are shown in Figure 7, using a total of 11,188 particles. When the displacement is loaded to 0.01 mm, significant tensile stress appears on the inside of the X-shaped crack, while obvious compressive stress exists on both sides of the crack. When the displacement is loaded to 0.02 mm, stress concentration occurs at the four tips of the crack, and there is a tendency to expand upwards and downwards. When the displacement is loaded to 0.03 mm, the stress concentration area continues to expand to both sides and tends to approach the middle, developing into a wing shape. Compared with the situation when the displacement is loaded to 0.02 mm, these areas are wider. When the displacement is loaded to 0.06 mm, the stress concentration area continues to increase, and larger compressive stresses appear on both sides of the crack. At this point, the stress concentration area no longer increases as the loading displacement increases, and the specimen fails.

According to the calculation results of example 3, shown in Figure 8, a total of 11,156 particles were used. When the displacement was loaded to 0.01 mm, significant tensile stress appeared on the upper side of the V-shaped crack, while there was obvious compressive stress between the two cracks. When the displacement was loaded to 0.02 mm, significant tensile stress appeared at the intersection of the two cracks. The tensile stress value and range outside the cracks were also large, and tended to expand upward and downward. When the displacement was loaded to 0.03 mm, the stress concentration area expanded to both sides, with an increased width. When the displacement was loaded to 0.06 mm, the length and width of the stress concentration area were larger than those at 0.03 mm and had a winged shape. At this point, as the loading continued, the area of the stress concentration zone did not increase significantly, and the specimen failed.

The calculation results for example 4 are shown in Figure 9, with a total of 23,987 particles used. When the displacement was loaded to 0.01 mm, there was a small tensile stress around the circular hole, while clear compressive stresses are present on the upper and lower sides of the hole. When the displacement is loaded to 0.02 mm, there is a significant concentration of tensile stress in the upper and lower regions of the hole. When the displacement is loaded to 0.03 mm, the region of stress concentration expands in both upward and downward directions, with an increase in width. When the displacement was loaded to 0.06 mm, the region experiencing stress concentration further enlarges. At this point, as the loading continues, the increase in the area of this region becomes less significant, and the specimen fails.

The calculation results of example 5 are shown in Figure 10, using a total of 23,811 particles. When the displacement was loaded to 0.01 mm, there was moderate tensile stress around the circular hole, and noticeable tensile stress occurred on the side of the inclined crack. When the displacement was loaded to 0.02 mm, the specimen mainly experienced compressive stress, with a significant tensile stress concentration at the bottom of the circular hole and on both sides of the crack. The magnitude and size of the tensile stress on both sides of the crack were significant. When the displacement was loaded to 0.03 mm, the stress concentration area expands in both upward and downward directions, and the width increases. When the displacement was loaded to 0.06 mm, the region experiencing stress concentration further expands and approaches the point of failure. At this point, as the loading continues, the area increase in this region became less significant, and the specimen failed.

From the above example, it can be observed that the specimen started to experience failure between 1000 and 2000 steps. The tensile stress before failure ranged from 0.2 to 0.5 MPa. The specimen experienced significant damage when loaded to 0.06 mm. Since each analysis step represents a loading of 10^−8^ m, every 1000 steps corresponds to a loading of 0.01 mm. Therefore, the specimen began to fail at a loading of 0.01–0.02 mm, and complete failure occurred at 0.06 mm. It can be concluded that different shapes of defects have a significant impact on the failure mode of rocks, while they have a minimal effect on the time it takes for the rock to fail.

### 4.3. Comparison with Existing Experimental or Numerical Simulation Results and Validation

In order to validate the accuracy of the proposed method in this paper, a comparison will be made between the results obtained in this study and the existing experimental and numerical simulation results [34]. The paper [35] presents numerical simulations of crack propagation in specimens with a single crack. The numerical simulation results, as depicted in Figure 11a, demonstrate that the specimens exhibit a wing-shaped crack expanding towards both sides after undergoing compression. The experimental results of the paper [36], as shown in Figure 11b, reveal the formation of four cracks. Among them, FS1 corresponds to a shear crack, while FT2, FT3, and FT4 represent tensile cracks. These results are comparable to the findings of example 2 in this study. The experimental results of the paper [2], as depicted in Figure 11c,d, illustrate the upward propagation of two main cracks originating from the tips of the V-shaped cracks. Additionally, the central portion of the V-shaped cracks extends downward until the specimen experiences ultimate failure. The aforementioned experimental and numerical results provide evidence for the accuracy of the numerical simulations in examples 1 to 3 of this paper.

The paper [37] elucidates the fracture mode of a compressively loaded specimen with an embedded circular aperture, as displayed in Figure 12a,b. Figure 12c illustrates the propagation of cracks during the uniaxial compressive failure of specimen A. The initial cracks, labeled as 1a and 1b, originate from the top and bottom of the circular aperture, respectively. On the other hand, Figure 12d portrays the uniaxial compressive failure of specimen A, showing the cracks initiating from the edges and propagating towards the top or bottom of the middle section. Both examples serve to validate the findings of this study. The aforementioned experimental results align closely with the outcomes presented in example 5 of this paper, thereby confirming the accuracy of the numerical simulations conducted in example 4 and example 5.

### 4.4. Influence of Particle Number

Figure 13a–c present the results of Case 1 calculations with particle numbers of 11,216, 15,101, and 19,937, respectively. By observing the figures, it can be concluded that the computational results exhibit good consistency across different particle numbers. It can be seen that the influence of particle number on the calculation results is relatively small. As the particle number increases, the stress distribution shown in the stress contour plot becomes clearer. However, when the number of particles is too large, the computation speed decreases. Therefore, it is necessary to determine the particle number reasonably during the calculation process.

## 5. The Relationship between Crack Propagation and Geometric Parameters of Cracks

### 5.1. The Relationship between Crack Propagation and Inclination Angle

The principal stress and crack propagation variation in rock under uniaxial compression with different crack inclinations is illustrated in Figure 14, where the angle between the crack and the horizontal direction ranges from 15° to 75°. From the graph, the following conclusions can be drawn: During the initial stage of compression, stress concentration occurs primarily on both sides of the crack. The crack propagation paths are similar, with crack propagation being perpendicular to the crack when the crack angle is small and closer to the crack angle as the crack angle increases. As the inclination angle increases, the extent of stress concentration gradually decreases, resulting in maximum tensile stress reduction. For smaller inclinations, the points of stress concentration are located near the ends of the crack rather than on the sides of the crack.

### 5.2. Relationship with Crack Width

The main stress and crack propagation of rock under uniaxial compression with different crack widths are shown in Figure 15, with crack widths of 0.6 mm, 1.2 mm, 1.8 mm, and 2.4 mm, respectively. It can be seen from the figure that with the increase in the crack width, the area of stress concentration becomes larger, but the variation is not significant; at the same time, the crack width has little effect on crack propagation.

### 5.3. Relationship with Crack Length

The principal stress and crack propagation variation in rock under uniaxial compression with different crack widths is depicted in Figure 16, considering crack lengths of 5 mm, 10 mm, 15 mm, 20 mm, and 25 mm, respectively. From the graph, it can be observed that as the crack length increases, the region of stress concentration expands, while the influence of crack width on crack propagation is relatively insignificant. For shorter crack lengths, the crack propagation path tends to be more direct and outward-oriented, whereas longer crack lengths result in more tortuous and inward-biased crack propagation paths. With an increasing crack length, the area of stress concentration gradually expands, leading to an increase in the maximum tensile stress magnitude.

## 6. Conclusions

This study used the SPH method to simulate the mechanical testing of several cracked specimens and compared the results with existing experimental and numerical simulation results. The tested specimens had different inclinations, widths, and lengths, and the relationships between crack propagation and crack inclination and width, and length were determined. The contributions of our study are as follows:a.This study applies the SPH method with a fifth-order smoothing function to solid mechanics. This method is relatively simple in regard to programming and can effectively simulate the crack propagation and failure processes of rock materials. Our numerical simulation results are in good agreement with previous experimental and numerical simulation results, demonstrating the method’s efficacy in studying crack propagation and providing guidance for rock mechanics research. The number of particles has a minor influence on the calculation results.b.Stress concentration often originates at the tip of a crack and tends to propagate longitudinally. For specimens with circular holes, crack propagation mainly occurs at the top and bottom of the hole. If there are cracks around the hole, the failure will preferentially occur at the crack rather than the top or bottom of the hole. The specimen starts to fail at a displacement of approximately 0.01 mm to 0.02 mm, and complete failure occurs at a displacement of around 0.06 mm. Different shapes of defects have a significant impact on the failure mode of rocks, while they have a minimal effect on the time it takes for the rock to fail.c.The lengths of pre-existing cracks and the angles between them and the horizontal direction have a significant impact on the crack propagation path and the size of the stress concentration area, while the width of reserved cracks has a relatively small effect on crack propagation. The larger the length of the crack and the smaller the angle between it and the horizontal direction, the larger the stress concentration area.

## Figures and Tables

**Figure 1 materials-16-06560-f001:**
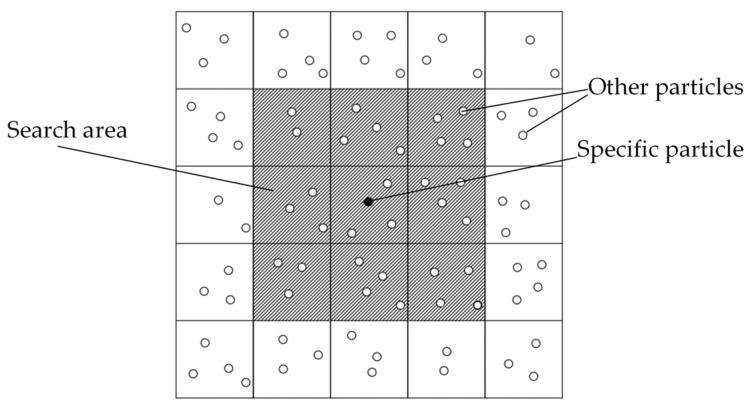
Search area of two-dimensional linked list search method.

**Figure 2 materials-16-06560-f002:**
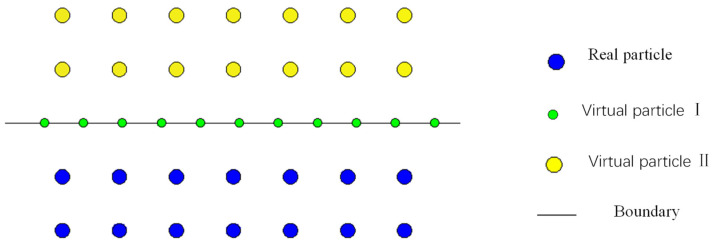
Boundary distribution map.

**Figure 3 materials-16-06560-f003:**
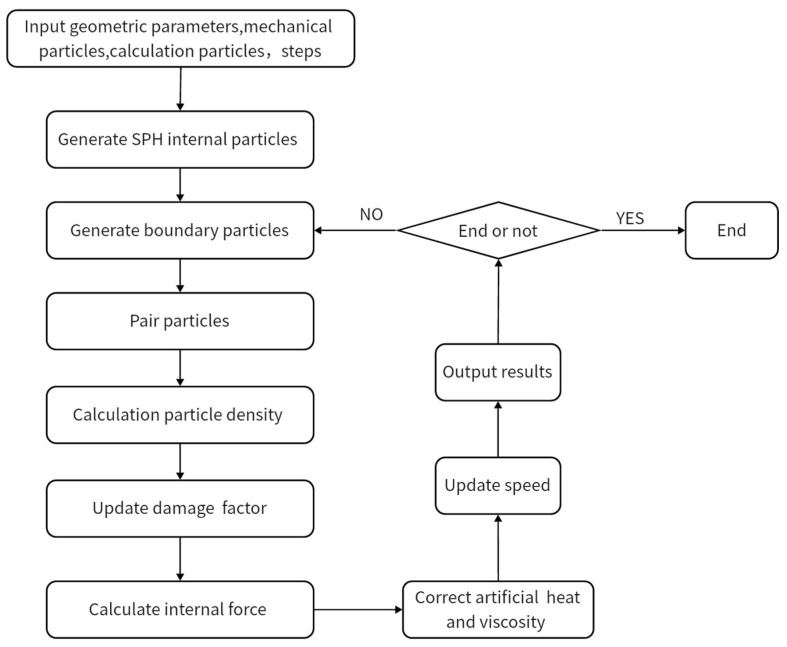
Program calculation process.

**Figure 4 materials-16-06560-f004:**
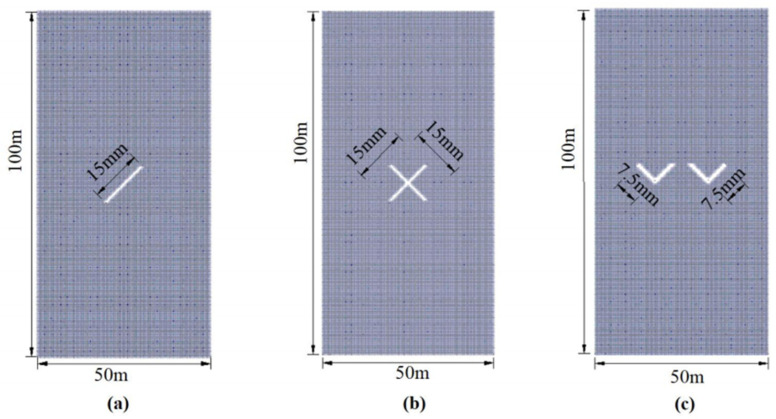
Model size and base particle distribution of example 1, example 2, and example 3. (**a**) The geometric dimensions of example 1. (**b**) The geometric dimensions of example 2. (**c**) The geometric dimensions of example 3.

**Figure 5 materials-16-06560-f005:**
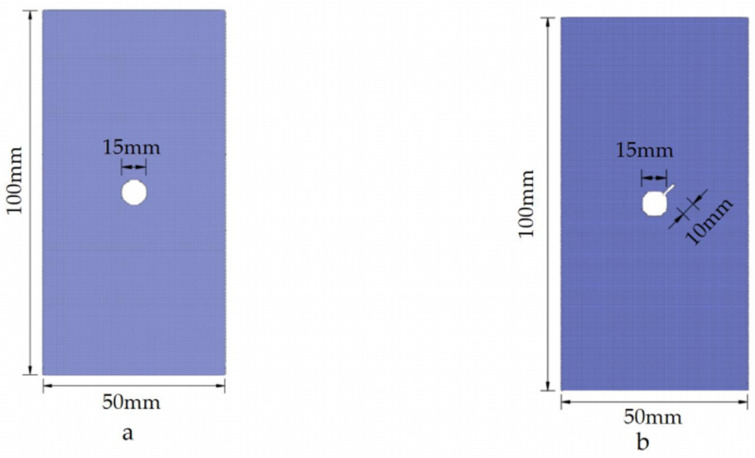
Model size and base particle distribution of example 4 and example 5. (**a**) The geometric dimensions of example 4. (**b**) The geometric dimensions of example 5.

**Figure 6 materials-16-06560-f006:**
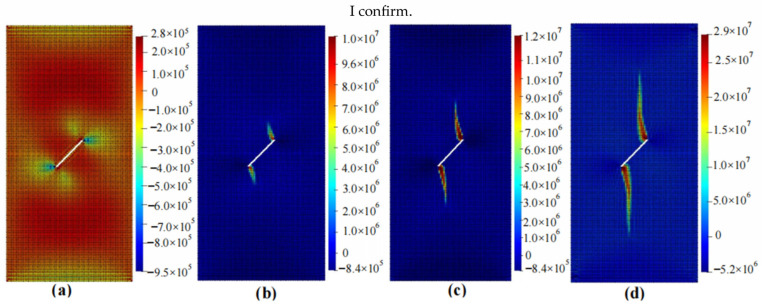
Maximum principal stress (Pa) and crack propagation process of example 1. (**a**) is the stress contour plot when loaded with a displacement of 0.01 mm. (**b**) is the stress contour plot when loaded with a displacement of 0.02 mm. (**c**) is the stress contour plot when loaded with a displacement of 0.03 mm. (**d**) is the stress contour plot when loaded with a displacement of 0.06 mm.

**Figure 7 materials-16-06560-f007:**
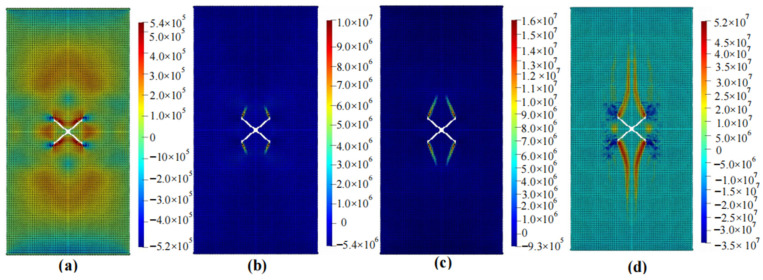
Maximum principal stress (Pa) and crack propagation process of example 2. (**a**) is the stress contour plot when loaded with a displacement of 0.01 mm. (**b**) is the stress contour plot when loaded with a displacement of 0.02 mm. (**c**) is the stress contour plot when loaded with a displacement of 0.03 mm. (**d**) is the stress contour plot when loaded with a displacement of 0.06 mm.

**Figure 8 materials-16-06560-f008:**
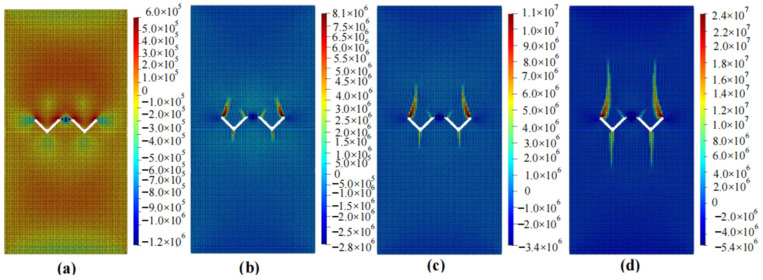
Maximum principal stress (Pa) and crack propagation process of example 3. (**a**) is the stress contour plot when loaded with a displacement of 0.01 mm. (**b**) is the stress contour plot when loaded with a displacement of 0.02 mm. (**c**) is the stress contour plot when loaded with a displacement of 0.03 mm. (**d**) is the stress contour plot when loaded with a displacement of 0.06 mm.

**Figure 9 materials-16-06560-f009:**
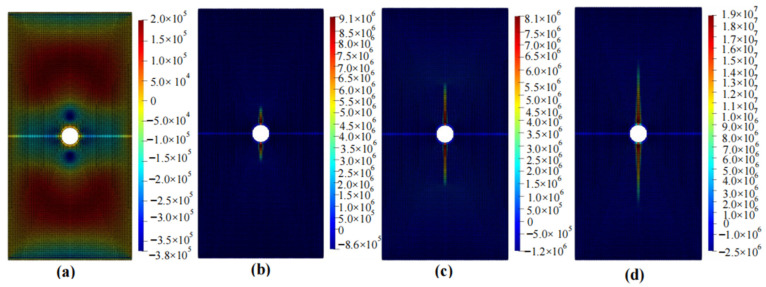
Maximum principal stress (Pa) and crack propagation process of example 4. (**a**) is the stress contour plot when loaded with a displacement of 0.01 mm. (**b**) is the stress contour plot when loaded with a displacement of 0.02 mm. (**c**) is the stress contour plot when loaded with a displacement of 0.03 mm. (**d**) is the stress contour plot when loaded with a displacement of 0.06 mm.

**Figure 10 materials-16-06560-f010:**
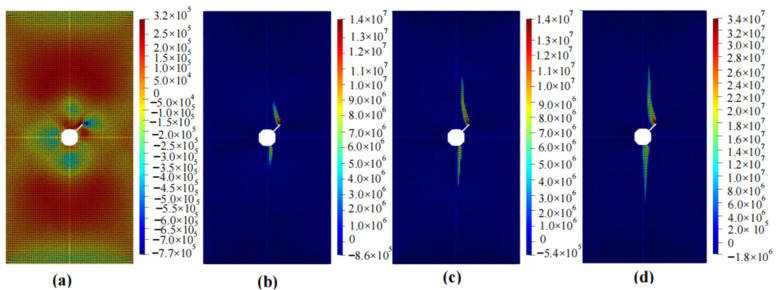
Maximum principal stress (Pa) and crack propagation process of example 5. (**a**) is the stress contour plot when loaded with a displacement of 0.01 mm. (**b**) is the stress contour plot when loaded with a displacement of 0.02 mm. (**c**) is the stress contour plot when loaded with a displacement of 0.03 mm. (**d**) is the stress contour plot when loaded with a displacement of 0.06 mm.

**Figure 11 materials-16-06560-f011:**
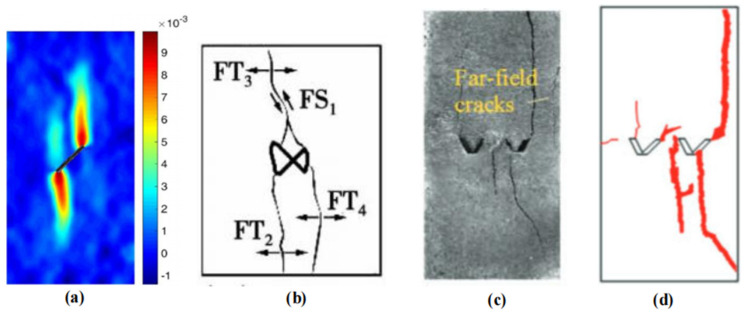
Simulation or experimental results from other papers. (**a**) represents the numerical simulation results from other papers. (**b**–**d**) represent the experimental results from other papers.

**Figure 12 materials-16-06560-f012:**
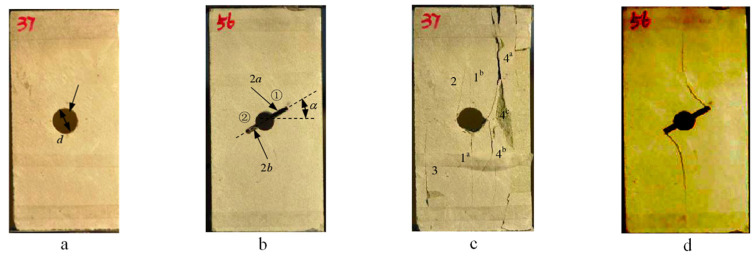
Simulation or experimental results from other papers. (**a**,**b**) represent the geometric dimensions of the specimens in other papers, while (**c**,**d**) represent the experimental results of these two specimens.

**Figure 13 materials-16-06560-f013:**
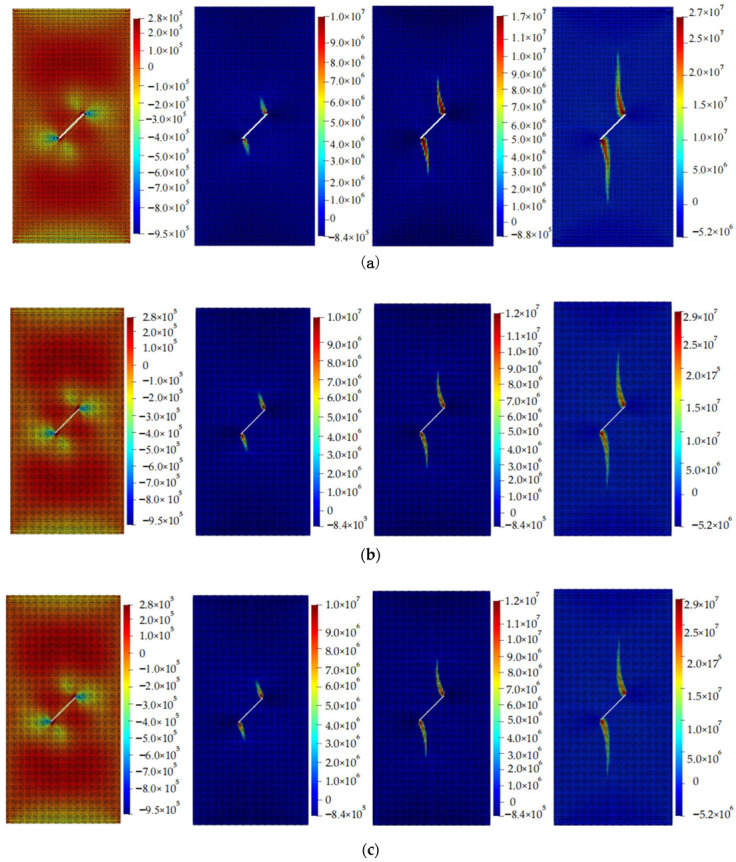
Crack propagation under different particle numbers (example 1). (**a**) is the stress contour plot when the number of particles is 11,216. (**b**) is the stress contour plot when the number of particles is 15,101. (**c**) is the stress contour plot when the number of particles is 19,937.

**Figure 14 materials-16-06560-f014:**
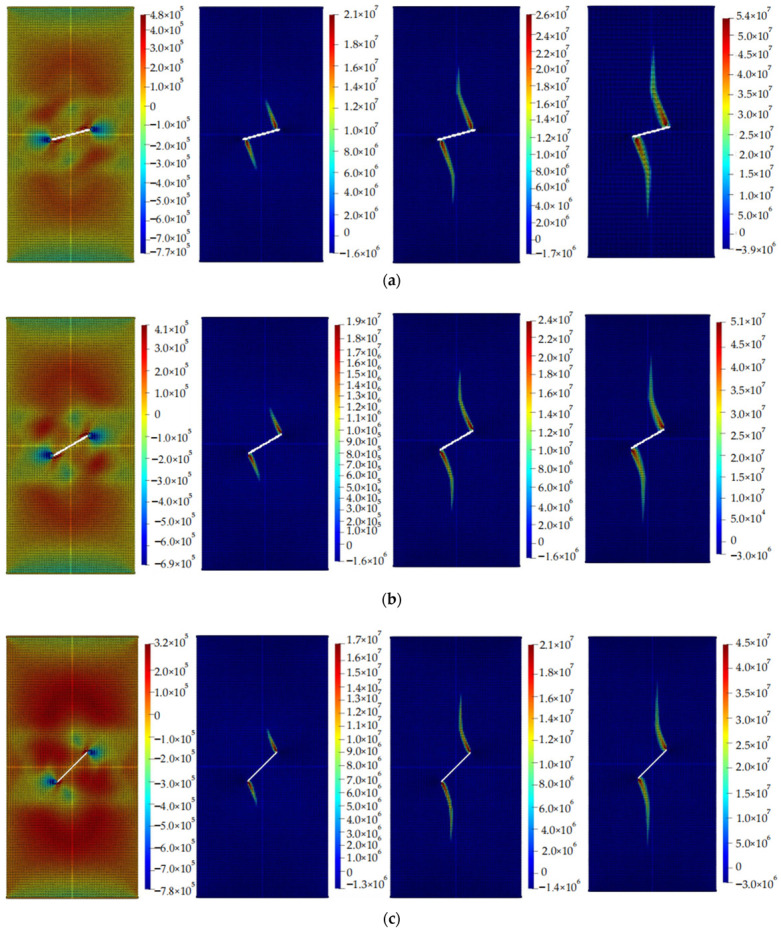
Crack growth diagram of rock under uniaxial compression with different crack inclination angles. (**a**) α = 15°; (**b**) α = 30°; (**c**) α = 45°; (**d**) α = 60°; (**e**) α = 75°.

**Figure 15 materials-16-06560-f015:**
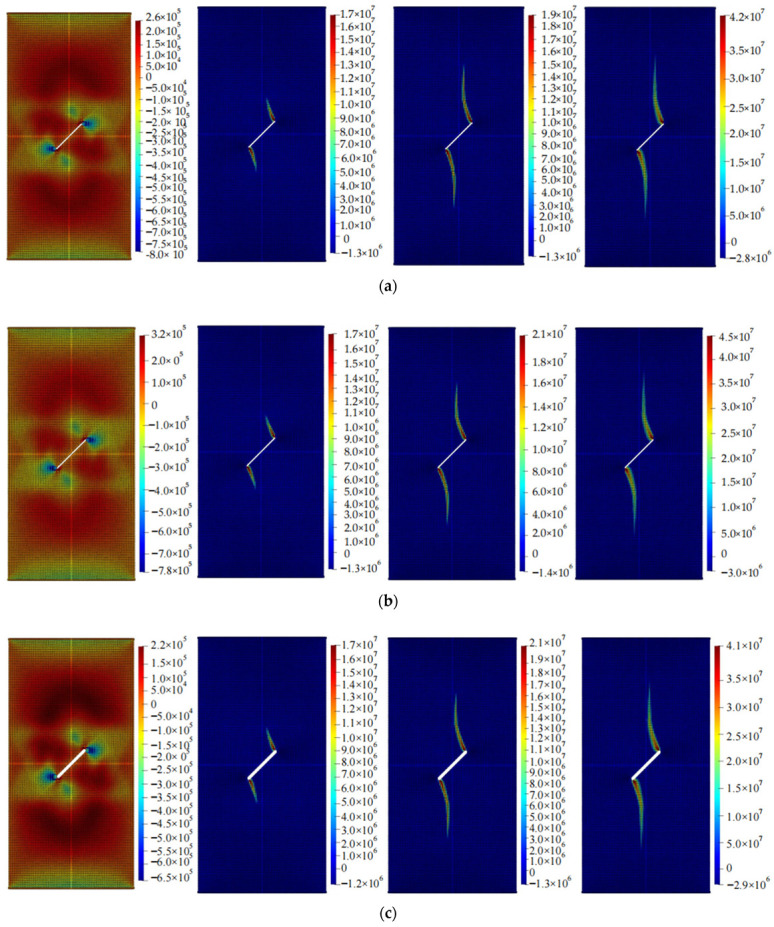
Crack growth diagram of rock under uniaxial compression with different crack widths. (**a**) Crack width 0.6 mm; (**b**) Crack width 1.2 mm; (**c**) Crack width 1.8 mm; (**d**) Crack width 2.4 mm.

**Figure 16 materials-16-06560-f016:**
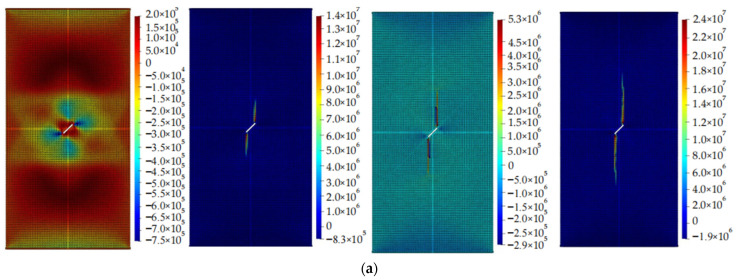
Crack growth diagram of rock under uniaxial compression with different crack lengths. (**a**) crack length 5 mm; (**b**) crack length 10 mm; (**c**) crack length 15 mm; (**d**) crack length 20 mm; (**e**) crack length 25 mm.

## Data Availability

The data used to support the findings of this study are included within the article.

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
