# Peer review of "Numerical Simulation Analysis of Fracture Propagation in Rock Based on Smooth Particle Hydrodynamics"

_materials, 2023, doi:10.3390/ma16196560_

Round 1

Reviewer 1 Report

1. Grammar and spelling checking:

- Lines 188, 224, 243, 256, and so on.

- Many sentences need to be double-checked for grammar and spelling corrections.

2. The numerical finite element (FE) crack propagation approach using local mesh refinement at the crack tip should be addressed in the Introduction (https://doi.org/10.1007/s00466-020-01863-9, https://doi.org/10.1016/j.engfracmech.2023.109304).

3. Was the Mohr-Coulomb criterion used for the fracture criteria? The author needs to address the original paper of the fracture model used in the paper. Moreover, the particle-based fracture description is not clear. Please add more explanation with figures.

4. The programming flowchart needs to be revised to be more clear.

5. In numerical examples, please compare the obtained results from this paper with other results to clearly demonstrate the effectiveness of the proposed method.

1. Grammar and spelling checking:

- Lines 188, 224, 243, 256, and so on.

- Many sentences need to be double-checked for grammar and spelling corrections.

- Missing ",", space at end of sentenses, ...

Author Response

Dear reviewer:

Thank you for your decision and constructive comments on my manuscript. We have carefully considered the suggestion of Reviewer and make some changes. We have tried our best to improve and made some changes in the manuscript.The following part that has been revised according to your comments.

Point 1: Grammar and spelling checking: Lines 188,224,243,256,and so on. Many sentences need to be double-checked for grammar and speling corrections.

Response 1: We apologize for the poor language of our manuscript. We worked on the manuscript for a long time and the repeated addition and removal of sentences and sections obviously led to poor readability. We have now worked on both language and readability and have also involved native English speakers for language corrections. We really hope that the flow and language level have been substantially improved.

Point 2: The numerical finite element (FE) crack propagation approach using local mesh refinement at the crack tip has been discussed in this paper.

Response 2: Thank you very much for the recommendation. This paper has already discussed the numerical finite element (FE) crack propagation approach using local mesh refinement atthe crack tip.

Point 3: Was the Mohr-Coulomb criterion used for the fracture criteria? The author needs to addressthe original paper of the fracture model used in the paper. Moreover, the particle-based fracturedescription is not clear Please add more explanation with figures.

Response 3: Thank you for your advice.By simultaneously applying the Mohr-Coulomb criterion and the maximum tensile stress criterion, reaching either one leads to failure. Once the particles reach the point of failure, the transmission of forces will cease.

Point 4: The programming flowchart needs to be revised to be more clear.

Response 4: Thank you for the suggestion. The flowchart has been revised.

Point 5:  In numerical examples, please compare the obtained results from this paper with other resultsto clearly demonstrate the effectiveness of the proposed method.

Response 5: Thank you very much for providing this highly convincing suggestion. The numerical examples in this paper are demonstrated, along with the comparison to the results of other studies, as shown in Section 4.3.

We appreciate your warm work earnestly and hope that the correction will meet with approval. Once again, thank you very much for your comments and suggestions.

Best regards.

Yours

sincerely.

Reviewer 2 Report

This presents.

The mechanical properties of rock-containing fractures have always been a focal point in the field of rock mechanics. Based on previous research, this paper proposes improvements to the sph method and applies it to the study of crack propagation in rocks containing fractures. The characteristics of crack propagation were obtained by conducting uniaxial compression tests and simulating crack propagation on various specimens with different shapes of cracks. The comparison between the simulated results in this study and the existing experimental or numerical simulation results confirms the validity of the Sph method employed in this paper. The present study utilizes the proposed methodology to analyze the influence of crack angle, width, and orientation on crack propagation. The results indicate that the sph method employed in this study effectively captures the crack propagation process in rock samples under uniaxial compression, providing valuable insights for the application of sph in engineering contexts.

This paper level is not enough for publication in this journal I can not recommend it for possible publication in this journal. However, I will encourage the authors to make the following changes and resubmit it.

1. The title is looking lengthy please make it concise if possible.

2. All figures quality is not good. Please make at least 300 dpi figures.

3. Author contribution is missing.

4. Mesh independent for simulation missing.

5. Validation of the simulation model is missing.

Author Response

Dear reviewer:

Thank you for your decision and constructive comments on my manuscript. We have carefully considered the suggestion of Reviewer and make some changes. We have tried our best to improve and made some changes in the manuscript. The following part that has been revised according to your comments.

Point 1: The title is looking lengthy please make it conctse if possible.

Response 1: Thank you for your suggestion. The title of this paper has been modified, changing it from " Research on the failure behavior of rock with cracks based on the Smoothed Particle Hydrodynamics" to " Numerical Simulation Analysis of Fracture Propagation in Rock Based on Smooth Particle Hydrodynamics".

Point 2: All figures quality is mot good. Pleas make at least 300 dpi figures.

Response 2: Thank you for your advice. The images in this paper have been modified and replaced with clearer ones.

Point 3: Author contribution is missing.

Response 3: Thank you for your suggestion. The author's contributions have been supplemented.

Point 4: Mesh independent for simulation missing.

Response 4: Thank you very much for your suggestion. he method used in this paper is the Smoothed Particle Hydrodynamics (SPH) method, which does not heavily rely on grids.

Point 5: Validation of the simulation model is missing.

Response 5: Thank you for your suggestion. The numerical examples in this paper are demonstrated, along with the comparison to the results of other studies, as shown in Section 4.3.

We appreciate your warm work earnestly and hope that the correction will meet with approval. Once again, thank you very much for your comments and suggestions.

Best regards.

Yours

sincerely.

Reviewer 3 Report

Dear authors,

I have reviewed the entitled manuscript materials-2616357 “Research on the failure behavior of rock with cracks based on 2 the Smoothed Particle Hydrodynamics”. There are fundamental drawbacks to developing results and comparing them that need to be addressed. Some of the major problems in this manuscript are listed below:

1- In the Abstract section, it is necessary to quantify the results.

2- In the introduction section, the literature review is well done. But the recent researches in the years 2020 to 2023 have been mentioned less. Authors please strengthen this section by adding new valid references.

Also, in the final part of the introduction, the authors did not mention their own work. At the end of the introduction, it is necessary to mention the objectives and benefits of the current research work.

3- Reference or references should be mentioned for the equations.

4- The different parts of Figures 13 to 15 are organized and presented on one page. Also, the quality of writing in Figures 6 to 15 is very poor and needs to be improved.

5- The conclusion is very qualitative and needs to be rewritten and strengthened.

Moderate editing of English language required

Author Response

Dear reviewer:

Thank you for your decision and constructive comments on my manuscript. We have carefully considered the suggestion of Reviewer and make some changes. We have tried our best to improve and made some changes in the manuscript. The following part that has been revised according to your comments.

Point 1: In the Abstract section it is necessary to quantify the results.

Response 1: Thank you for your advice. The abstract section has been revised, and quantitative results have been added.

Point 2: In the introduction section. the literature review is well done. But the recent researches in theyears 2020 to 2023 have been mentioned less. Authors please strengthen this section by addingnew valid references. Also, in the final part of the introduction, the authors did not mention their own work. At the end ofthe introduction, it is necessary to mention the objectives and benefits of the current researchwork.

Response 2: Thank you for your suggestion. The papers from 2020 to 2023 have been cited, and the conclusion has also been modified to highlight the benefits of this research work.

Point 3: Reference or references should be mentioned for the equations.

Response 3: Thank you very much for your suggestion.

Point 4: The different parts of Figures 13 to 15 are organized and presented on one page. Also, thequality of writing in Figures 6 to 15 is very poor and needs to be improved.

Response 4: Thank you very much .The references for the equations have been cited.

Point 5: The conclusion is very qualitative and needs to be rewriten and strengthened.

Response 5: Thank you for your suggestion. The conclusion has been rewritten.

Language:We apologize for the poor language of our manuscript. We worked on the manuscript for a long time and the repeated addition and removal of sentences and sections obviously led to poor readability. We have now worked on both language and readability and have also involved native English speakers for language corrections. We really hope that the flow and language level have been substantially improved.

We appreciate your warm work earnestly and hope that the correction will meet with approval. Once again, thank you very much for your comments and suggestions.

Best regards.

Yours

sincerely.

Round 2

Reviewer 1 Report

Thank you for revising the manuscript. I think the authors have put a lot of efforts into the revised version. However, some points need to be pointed out.

1. In Abstract, "The results indicate that the specimens start to fail when the displacement is loaded from 0.01mm to 0.02mm, and complete failure occurs at 0.06mm, demonstrating a wing-shaped crack pattern" is not clear for the reader. I think it should be placed in the numerical results.

2. "Smooth Particle Hydrodynamics (SPH)" should be mentioned 1 time, the shorten term "SPH" can then be used afterward. Please check it carefully.

3. Line 109: Why is every 1st character in UPPER form, while others are not? Check the 1st character in lines 255, 267,...

4. The math symbols should be inclined with texts, see line 119-121, 127, 133,... Please use bold symbols to distinguish the coordinate vector and matrix from scalar terms.

5. Why did all specimens have the same crack propagation behavior: the specimens start to fail when the displacement is loaded from 0.01mm to 0.02mm, and complete failure occurs at 0.06mm?

Please carefully check all the phrases and sentences in the manuscript. There are still a lot of grammar and spelling mistakes in the revised version. Thank you.

Author Response

Dear reviewer:

Thank you again for taking your time to review this manuscript. Please find my itemized responses in below and my revisions in the re-submitted files.

Point 1: In Abstract"The results indicate that the specimens start to fail when the displacement isloaded from 0.01mm to 0.02mm, and complete failure occurs at 0.06mm, demonstrating a wingshaped crack pattern" is not clear for the reader. I think it should be placed in the numericalresults.

Response 1: Thank you for your advice. This sentence has been removed from the abstract.

Point 2: "Smooth Particle Hvdrodynamics (SPH) should be mentioned 1 time. the shorten term "SPH"can then be used afterward. Please check it carefully.

Response 2: Thank you for the suggestion. After conducting a thorough check, it has been determined that other sections of the paper have been revised.

Point 3: Line 109: Why is every 1st character in UPPER form, while others are not? Check the 1stcharacter in lines 255267...

Response 3: Thank you very much for the recommendation. The characters have been capitalized.

Point 4: The math symbols should be inclined with texts, see line 119-121, 127, 133.... Please use boldsymbols to distinquish the coordinate vector and matrix from scalar terms.

Response 4: Thank you for your advice. The symbols have been modified .

Point 5: Why did all specimens have the same crack propagation behavior: the specimens start to fail when the displacement is loaded from 0.01mm to 0.02mm, and complete failure occurs at0.06mm?

Response 5: Thank you for the question. Since the dimensions, parameters, and loading methods of each case in this study are identical, except for slight variations in crack size, the impact of dimension differences is not significant. Different shapes of defects have a significant impact on the failure mode of rocks, while they have a minimal effect on the time of rock failure.

Point 6: Please carefully check all the phrases and sentences in the manuscrpt. There are still a lot ofgrammar and spelling mistakes in the revised version. Thank you.

Response 6: We apologize for the poor language of our manuscript. We have checked for grammatical errors and made the necessary modifications.

We appreciate your warm work earnestly and hope that the correction will meet with approval. Once again, thank you very much for your comments and suggestions.

Best regards.

Yours

sincerely.

Reviewer 2 Report

Although most of the concerns are resolved in the updated version. However, the following two points still need to be addressed.

1. Mesh independent for simulation missing. If you are not doing this, how do you ensure the accuracy of results?

2. Validation of the simulation model is missing. It is mentioned that validation is done with literature, but I could not find it in the paper. The validation must be both a comparison with literature results qualitatively (surface plots) and Quantitative (Line plot) to prove your simulation model is working correctly.

3. For the above points you can see the following research article and its supplementary file. Do a similar representation in your case and cite this paper as proof.

-Numerical analysis of non-aligned inputs M-type micromixers with different shaped obstacles for biomedical applications

Author Response

Dear reviewer:

Thank you again for taking your time to review this manuscript. Please find my itemized responses in below and my revisions in the re-submitted files.

Point 1: Mesh independent for simulation missing. lf you are not doing this, how do you ensure theaccuracy of results?

Response 1: Thank you very much for providing this high question. The SPH method is based on discretizing continuous materials into a series of basic units called "particles", each with its own mass, position, velocity and other physical properties. These particles interact with each other through local interpolation and smoothing kernel functions. Although no mesh is used, accurate results can still be obtained because forces can be transmitted through the interaction forces between particles. Section 3.4 of this paper discusses the study of crack propagation process in example 1 for different particle numbers. The results indicate that the computational results exhibit good consistency across different particle numbers.

Point 2: Validation of the simulation model is missing. It is mentioned that validation is done with literature, but I could not find it in the paper. The validation must be both a comparison with literature results qualitatively (surface plots) and Quantitative (Line plot) to prove your simulationmodel is working correctly.

Response 2: Thank you for the suggestion. In Section 3.3 of this paper, a comparison is made between the simulation results of this study and those from other papers. Figure 11(a) validates Case 1, Figure 11(b) validates Example 2, Figures 11(c) and 11(d) validate Example 3, while Figures 12(a) and 12(c) validate Example 4, and Figures 12(b) and 12(d) validate Example 5

Point 3: For the above points you can see the following research article and its supplementary file. Doa similar representation in your Example and cite this paper as proof.

Response 3: Thank you very much for the recommendation. This paper is highly informative and has been cited for reference.

We appreciate your warm work earnestly and hope that the correction will meet with approval. Once again, thank you very much for your comments and suggestions.

Best regards.

Yours

sincerely.

Reviewer 3 Report

Dear Authors

The manuscript-R1 has been significantly improved. Please change section "5-Result" to "5-Conclusion". After revising this, the article can be published.

Moderate editing of English language required

Author Response

(The authors gave the same response as above.)
